# Cross-Cultural Adaptation and Psychometric Properties of the Traditional Chinese Version of the Italian Spine Youth Quality of Life (ISYQOL) Questionnaire

**DOI:** 10.3390/healthcare11192683

**Published:** 2023-10-05

**Authors:** Ava Ying Cheng, Pik Kwan Jim, Ning Wai Kwan, Stephen W. W. Chan, Jason P. Y. Cheung, Prudence W. H. Cheung, Stefano Negrini, Chelsia K. C. Cheung, Arnold Y. L. Wong, Eric C. Parent

**Affiliations:** 1Department of Rehabilitation Sciences, The Hong Kong Polytechnic University, Hong Kong SAR, China; ying-ava.cheng@connect.polyu.hk (A.Y.C.); 20114318g@connect.polyu.hk (P.K.J.); 20114226g@connect.polyu.hk (N.W.K.); ckaching.cheung@polyu.edu.hk (C.K.C.C.); 2Allied Health Department (Physiotherapy), Hong Kong Children’s Hospital, Hong Kong SAR, China; ccwwz01@ha.org.hk; 3Department of Orthopaedics and Traumatology, The University of Hong Kong, Hong Kong SAR, China; cheungjp@hku.hk (J.P.Y.C.); gnuehcp6@hku.hk (P.W.H.C.); 4Department of Biomedical Surgical and Dental Sciences, University “La Statale”, 20122 Milan, Italy; stefano.negrini@unimi.it; 5IRCCS Istituto Ortopedico Galeazzi, 20161 Milan, Italy; 6Department of Physical Therapy, University of Alberta, Edmonton, AB T6G 2G4, Canada; eparent@ualberta.ca

**Keywords:** scoliosis, health-related quality of life, Chinese version of Italian Spine Youth Quality of Life, Scoliosis Research Society-22 revised questionnaire

## Abstract

The Italian Spine Youth Quality of Life (ISYQOL) is a validated health-related quality of life (HRQOL) questionnaire for teenagers with adolescent idiopathic scoliosis (AIS). We culturally-adapted ISYQOL to traditional Chinese (ISYQOL-TC) and then recruited 133 conservatively treated teenagers with AIS to complete the ISYQOL-TC and the Chinese version of the Scoliosis Research Society-22 revised (SRS-22r) questionnaire, nine-item Patient Health Questionnaire (PHQ-9), seven-item Generalized Anxiety Disorder scale (GAD-7), and numeric pain rating scale (NPRS). They repeated ISYQOL-TC two weeks later. The internal consistency, unidimensionality, and test–retest reliability were measured using the Cronbach’s alpha, Rasch measurement models, and intra-class correlation coefficients (ICC3,1), respectively. The concurrent validity of the ISYQOL-TC with SRS-22r, and its construct validity with other questionnaires were evaluated using Spearman correlation coefficients. The ISYQOL-TC demonstrated good internal consistency (Cronbach’s alpha 0.90 and 0.89 for items 1–13 and items 1–20), and excellent test–retest reliability (ICC3,1 = 0.95–0.96). The Rasch analysis supported the unidimensionality of all 20 items in ISYQOL-TC. The ISYQOL-TC percentage scores were positively correlated with SRS-22r total scores (r = 0.65; *p* < 0.05), but were negatively related to PHQ-9, GAD-7, and NPRS scores (r = −0.46 to −0.39; *p* < 0.01). Collectively, the ISYQOL-TC is a reliable and valid instrument for evaluating HRQOL in Chinese teenagers with AIS.

## 1. Introduction

Adolescent idiopathic scoliosis (AIS) is the most common three-dimensional structural spine change among teenagers aged between 10 and 18 years. The reported prevalence of AIS in teenagers ranges from 0.47% to 5.2% globally, and is 2.4% in Eastern China [1,2,3,4]. Girls are 1.5 to 3 times more likely than boys to have AIS. The spinal curvature, as measured by the Cobb method, substantially increases with age [2,4]. Because uncontrolled AIS may lead to pain, curve progression, and physical and psychological dysfunctions in patients [5,6], patients with AIS may have suboptimal physical and psychological wellbeing, and/or health-related quality of life (HRQOL) [7,8].

The Scoliosis Research Society-22 questionnaire revised (SRS-22r) was developed to assess the HRQOL in patients with idiopathic scoliosis (including AIS and hypokyphosis) with or without treatments. SRS-22r has been translated into different languages, and it is the most widely used validated instrument worldwide [9]. Despite its excellent internal consistency (Cronbach’s alpha = 0.75–0.92) and test–retest reliability (intraclass correlation coefficient > 0.8), SRS-22r is limited by high ceiling effects (56.9%) [10,11]. Research has shown that SRS-22r had ceiling effects of up to 52.9% in the functional domain, and 67% in the pain domain [12,13,14].

The Italian Spine Youth Quality of Life questionnaire (ISYQOL) is a new alternative for evaluating the impacts of AIS on the HRQOL of patients with AIS [14,15,16]. ISYQOL comprises 20 items with the last 7 items tailored for patients with bracing. Using the Rasch analysis, the ordinal scale of ISYQOL is converted to an interval scale presented as a percentage ranging from 0% to 100%. People with and without a brace need to answer 20 and 13 questions, respectively. Researchers and clinicians can directly compare the ISYQOL results between bracers and non-bracers. Importantly, ISYQOL has demonstrated a smaller ceiling effect (<2.5%) for assessing the HRQOL of patients with AIS [14,15,16], as well as good internal consistency (Cronbach’s alpha > 0.7) and test–retest reliability (ICC > 0.90) in various translated versions [14,15,16,17].

The original Italian questionnaire has been translated and cross-culturally validated in different languages (e.g., Arabic, Canadian-French, Chinese, English, and simplified Chinese [13,14,16,17,18,19]). Although the simplified Chinese version has demonstrated good reliability and validity, the perceived meaning of wording by simplified Chinese users differs from traditional Chinese users due to cultural differences [20]. Therefore, it is essential to cross-culturally adapt and validate the ISYQOL into traditional Chinese to evaluate the HRQOL of this subgroup of patients.

Considering the aforementioned background, this study aimed to (1) translate and cross-culturally adapt the original ISYQOL into the traditional Chinese version; and (2) evaluate the internal consistency, structural and construct validity, as well as the test–retest reliability of the traditional Chinese version of the ISYQOL (ISYQOL-TC) among patients with AIS in Hong Kong.

## 2. Materials and Methods

This study was approved by the institutional review board at The University of Hong Kong and The Hong Kong Polytechnic University.

### 2.1. Study Design

The study consisted of two phases. Phase 1 involved the translation of the original ISYQOL into traditional Chinese. Phase 2 investigated the internal consistency, test–retest reliability, structural validity, and construct validity (convergent and divergent) of the ISYQOL-TC.

### 2.2. Participants

Teenagers who spoke Cantonese and read traditional Chinese, aged between 10 and 18 years, and were diagnosed with AIS by physicians were recruited from the scoliosis clinic in the Duchess of Kent Children’s Hospital by convenient sampling between January and June 2021 (Figure 1). Participants were required to have a thoracic (T3–L3) and/or lumbar curve with coronal curve angles between 11° and 60° on an anteroposterior radiograph taken within the last 6 months. They should have been treated conservatively or be waiting for spine surgery. Teenagers were excluded if they had (1) a history of spine surgery, spinal fracture, or trunk or lower limb trauma; (2) neurological conditions unrelated to scoliosis; (3) comorbidities unrelated to scoliosis that affect HRQOL; or (4) difficulty in understanding written Chinese. Parental consent and child assent were obtained in written form prior to data collection.

For the phase 1 study, six patients with AIS participated in individual cognitive debriefing interviews to provide feedback regarding the readability/appropriateness of the ISYQOL-TC, as suggested by the Function Assessment of Chronic Illness Therapy (FACIT) translation and linguistic validation methodology [21].

For the phase 2 study, the sample size for the validation study was determined based on four factors. First, the Consensus-based Standards for the Selection of Health Measurement Instruments (COSMIN) recommends 50 and 99 participants for evaluating internal consistency and content validity of a questionnaire [22]. Second, the COSMIN recommends at least 100 participants for assessing structural validity or test–retest reliability [22]. Third, previous Rasch analysis research showed that a sample size over 100 obtained fewer incorrectly ordered items in a questionnaire [23]. Fourth, the estimated sample size for the convergent validity testing was 80 if the statistical power was set at 80%, with the alpha level at 0.05 to detect a hypothesized correlation of 0.5 [24]. Given the above, the current validation study recruited at least 100 participants with AIS.

### 2.3. Phase 1: Translation and Cross-Cultural Adaptation of Questionnaire

The translation process followed the FACIT translation and linguistic validation method [21]. This double-back-translation method is more rigorous than a single translation and translation by a committee. Specifically, two professional bilingual (Italian and Cantonese) translators independently forward translated the original ISYQOL into ISYQOL-TC based on the semantic meaning of each item. Another bilingual reconciler compared and resolved discrepancies between both versions. An expert panel comprised an orthopedist, a pediatric physiotherapist, and a bilingual translator reviewed the translation processes, and verified the conceptual, idiomatic, and semantic equivalence of ISYQOL-TC. Another bilingual translator, who was blinded to the original ISYQOL and any of the forward translated versions, back translated the ISYQOL-TC into Italian. The back-translated version was then sent to the developer of ISYQOL (SN) for comment. SN’s feedback was then used to finalize the ISYQOL-TC by the panel. A proofreader then proofread the questionnaire. Six female teenagers with AIS were then recruited from the scoliosis clinic for cognitive debriefing interviews. They were asked to share their perceived meaning of each item in the ISYQOL-TC, and to give their rationale for answering each item. If any teenagers had difficulty in understanding a given item, the item would be revisited by the expert panel for potential modification.

### 2.4. Phase 2: Evaluations of Psychometric Properties of the ISYQOL-TC

This phase collected data from children with AIS attending the scoliosis clinic. Participants aged under 16 years required both written parental consent and child assent. Participants aged 16 years or older provided their written consent. Participants completed a demographic questionnaire regarding sex, age, the current AIS treatment, years of treatment, plan for surgery, phone number, and email address. They also completed the ISYQOL-TC, a SRS-22r, and five Chinese questionnaires (Patient Health Questionnaire-9 (PHQ-9), General Anxiety Disorder-7 (GAD-7), The Mastery Scale (MAS), 11-point Numeric Pain Rating Scale (NPRS), and the 20-item Simplified Coping Style Questionnaire (SCSQ)). To evaluate the reliability of ISYQOL-TC, participants were instructed to complete an online version of ISYQOL-TC after 14 days. A maximum of three phone calls were made to remind participants.

#### 2.4.1. ISYQOL

The 20-item ISYQOL questionnaire is the first Rasch-consistent questionnaire developed for assessing the HRQOL of teenagers with spinal deformities [8,25]. It comprises two domains: 13-item spinal health; and 7-item impacts of bracing (Table A1). Questions evaluating the presence of spine-related problems were coded as 0 (never), 1 (sometimes), and 2 (often). Questions investigating the presence of positive thoughts were coded as 0 (often), 1 (sometimes), and 2 (never). The total scores of ISYQOL are 26 and 40 for the unbraced and braced versions, respectively [8]. The ordinal scale score is converted to an interval scale score presented as a percentage ranging from 0 to 100. Participants with and without a brace answered 20 and 13 questions, respectively. Higher percentages indicate better HRQOL. ISYQOL has demonstrated high internal consistency (Cronbach’s alpha > 0.80) and test–retest reliability (intraclass correlation coefficient, ICC > 0.9) in Canadian-French, Persian, Polish, and simplified Chinese versions for conservatively treated patients [8,14,18,26].

#### 2.4.2. Chinese Version of SRS-22r

The original SRS-22r questionnaire was developed based on the classical test theory and has shown good psychometric properties [10]. It contains 5 domains: function (5 items), pain (5 items), self-image (5 items), mental health (5 items), and satisfaction with management (2 items) [10]. Each item has five answers with scores ranging from 1 to 5. Higher scores illustrate better HRQOL. The traditional Chinese version of SRS-22r has demonstrated high internal consistency (Cronbach’s alpha > 0.75 in function, pain, and mental health domains) and concurrent validity with the 36-Item Short Form Survey (correlation coefficient: r = 0.77, *p* < 0.01) in the function domain) [27].

#### 2.4.3. Other Chinese Questionnaires

The PHQ-9 is a commonly used questionnaire for screening, monitoring, and measuring the severity of depression in the past two weeks among teenagers aged between 13 and 17 years old [28]. The GAD-7 was used for screening and assessing the severity of generalized anxiety disorders [29]. The MAS evaluates individuals’ perceived mastery of their own lives [30,31]. The 11-point NPRS is a unidimensional measure of pain, ranging from 0 (no pain) to 10 (the worst imaginable pain). An individual’s coping style preference to deal with stress was quantified by SCSQ [32,33]. The SCSQ has an active coping style dimension (SCSQ-A) and a passive coping style dimension (SCSQ-P) [32]. The details of these questionnaires are delineated in Table A2.

### 2.5. Data Analysis

#### 2.5.1. Semantic Equivalence Score

The expert panel evaluated the semantic equivalence of each item of the ISYQOL-TC on a 4-point Likert scale (1, 2, 3, and 4 indicate not equivalent, somewhat equivalent, quite equivalent, and highly equivalent, respectively). Semantic equivalence means the proportion of items in a questionnaire that is rated as 3 or above [34].

#### 2.5.2. Content Validity Index (CVI)

The CVI indicates the content similarities between the translated items and the original items [35]. Each item was graded on a 4-point Likert scale (1 = ”not relevant”, 4 = ”very relevant”). The content validity index indicates the proportion of items being rated as 3 or above in a given questionnaire [34]. An index of 0.8 or above is deemed to have good semantic equivalence or content validity [36].

#### 2.5.3. Psychometric Property Analysis

The internal consistency, test–retest reliability, and construct validity of ISYQOL-TC were analyzed using SPSS v25.0 (IBM Corp., Armonk, NY, USA). The structural validity was assessed via the Rasch analysis of unidimensionality using the WINSTEPS Rasch software 4.0.1 [37].

##### Internal Consistency

Internal consistency evaluates the degree of interrelatedness among different items on the same test/scale. It determines whether the items investigate the same construct. The Cronbach’s alpha and item-total correlation of the ISYQOL-TC items were calculated to determine the internal consistency of each domain. Cronbach’s alpha values greater than 0.7 indicate good internal consistency [38,39].

##### Test–Retest Reliability

The test–retest reliability of the ISYQOL-TC scores between 14 days was evaluated using the ICC3,1 model. This duration was chosen because the spinal condition should be stable, but participants probably could not remember their previous rating.

##### Standard Error of Measurement (SEM) and 95% Confidence Minimal Detectable Change (MDC95)

SEM estimated the distribution of a person’s “true” score for repeating the same test. It is calculated by $SEM=standard\;deviation\times \sqrt{\left(1-ICC\right)}\;$ [40]. MDC95 means the smallest magnitude of true change between two repeated tests that is observed with a 95% level of confidence. It was calculated as $=\pm 1.96\times SEM\sqrt{2}$ [41].

##### Structural Validity

The unidimensionality of items in the ISYQOL-TC was evaluated using the Rasch measurement model [42]. Specifically, a Principal Component Analysis (PCA) of residuals was performed. Unidimensionality was confirmed if the “raw variance explained by the measures” accounted for >40% of the total variance [43,44], the eigenvalue of the second largest component was <3 [37], and the ratio of the “first component dimension” to the “largest secondary component” eigenvalue was >4 [45]. Additionally, the item fit statistics were conducted to determine how each item’s fit matches with the expected hierarchy of difficulty within a given domain in the Rasch model [42]. Both infit and outfit statistics were analyzed to determine how well the items fit the construct. The item fit statistics should be between 0.5 and 1.5 [46].

##### Convergent and Divergent Validity

Spearman correlation coefficients were used to examine the association between the percentage measure of ISYQOL-TC and various questionnaire scores. Questionnaires measuring similar constructs (i.e., SRS-22r, PHQ-9, GAD-7, MAS, or NPRS) should have moderate (ρ > |0.3|) to high correlation (ρ > |0.5|) [47]. Conversely, questionnaires (e.g., SCSQ) of dissimilar construct should show a negligible/low correlation (divergent validity).

## 3. Results

### 3.1. Demographics and Clinical Data

Ninety-nine female and thirty-four male teenagers (mean age: 13.9 ± 2.1 years) with an average coronal curve degree of 27° were recruited (Table 1). Approximately 43% of participants were wearing braces.

### 3.2. Semantic Equivalence and Content Equivalence

Both the semantic equivalence score and CVI revealed that 5% and 95% of items had scores of three and four, respectively (Table A3). It supported that all items in ISYQOL-TC were relevant to the corresponding constructs of the ISYQOL-TC, and the translated items showed good content validity [41].

### 3.3. Cognitive Debriefing Interviews

Six participants with AIS (three with and three without bracing, mean age 12.2 ± 1.8 years) completed the ISYQOL-TC and underwent individual interviews. All participants expressed that they understood all items in the questionnaire and could provide justifications for their selected answers. They deemed that all items were relevant to their condition, and that the questionnaire was easy to complete. The findings indicated no need to make any changes to the items in the ISYQOL-TC.

### 3.4. Internal Consistency and Test–Retest Reliability

All domains in ISYQOL showed good internal consistency (Table 2). The Cronbach’s alpha values of the spine health and brace domains were 0.89 and 0.79, respectively. Furthermore, the Cronbach’s alpha values from participants with (answered 20 items) and without a brace (answered 13 items) were 0.89 and 0.90, respectively (Table 2). The item-total correlations ranged from 0.27 to 0.76 (Table A4, Table A5, Table A6 and Table A7).

Almost all (99%, n = 131) participants completed the test and retest and their mean scores were similar (Table A8). The ICC3,1 was 0.95 for the spine health domain, and 0.96 for the brace domain. The ICC3,1 values for participants with (answered 20 items) and without bracing (answered 13 items) were 0.96 and 0.95, respectively (Table 2).

### 3.5. SEM and MDC95

The SEMs of ISYQOL-TC were 4% and 2% for the unbraced and braced groups, respectively. The MDC95 was 12% for the unbraced group and 6% for the braced group (Table 2).

### 3.6. Ceiling Effect

No participant scored maximum in the two domains or the total score of ISYQOL-TC, which represents 0% of the ceiling effect. For SRS-22r, 2% of the ceiling effect was observed in the total scores, ranging from 3% to 51% for the five domains (Table A9).

### 3.7. Structural Validity

The total variance explained by the Rasch-derived measure of all 20 items was 45.9% with an eigenvalue of 17 (Table A10). The Rasch PCA results showed that the first contrast of residuals (second major component) of all 20 items was 2.5. The eigenvalue ratio of “first component dimension” to the “second major component” was 6.8. These indicated that the ISYQOL-TC demonstrated unidimensionality in all 20 items. The fit statistics test revealed that all items showed acceptable levels in both infit and outfit, except for item 13 (Table A10). Although the outfit value of item 13 exceeded 1.5, it remained within a reasonable range (<2.0) that did not distort nor degrade the measurement.

### 3.8. Construct Validity

To compare the scores of participants with (answered 20 items) and without bracing (answered 13 items), the ISYQOL raw scores were converted to interval scores. The mean and standard deviation of the ISYQOL-TC raw scores and interval scores, as well as SRS-22r scores are presented in Table 3. The ISYQOL-TC interval scores were moderately related to the SRS-22r scores (r = 0.65, *p* < 0.01) (Table 3).

As hypothesized, the ISYQOL-TC interval scores displayed moderate negative correlations with PHQ-9, GAD-7, and NPRS (*p* < 0.01), while ISYQOL-TC interval scores showed moderate positive correlations with MAS (*p* < 0.01) (Table 4). No significant correlation was found between ISYQOL-TC interval scores and SCSQ-A or SCSQ-P scores (Table 4).

## 4. Discussion

The current study adopted the FACIT translation methodology to translate and culturally adapt the original ISYQOL into traditional Chinese [48] to ensure the cultural relevance of wordings, and to minimize deviations from the Italian wording or structure. Our cognitive debriefing interviews revealed that it was unnecessary to change the wordings of ISYQOL-TC because teenagers understood the translated questionnaire well. The ISYQOL-TC demonstrated good internal consistency, test–retest reliability, and unidimensionality. The moderate positive associations between ISYQOL-TC interval scores and SRS-22r scores substantiated that both scales had a very similar construct. Conversely, the ISYQOL-TC interval scores were weakly and negatively correlated with PHQ-9, GAD-7, and NPRS (*p* < 0.01), while ISYQOL-TC interval scores had a weak positive correlation with MAS scores (*p* < 0.01). Further, ISYQOL-TC interval scores were unrelated to SCSQ-A or SCSQ-P scores.

### 4.1. Good Internal Consistency, Test–Retest Reliability, and Unidimensionality

The ISYQOL-TC exhibited good internal consistency and test–retest reliability. Our study revealed that the ISYQOL-TC had good internal consistency in measuring HRQOL in AIS participants with and without bracing. These results concurred with the other translated versions of ISYQOL (e.g., simplified Chinese and English versions). Liu et al. used the simplified Chinese version of ISYQOL to assess the HRQOL of patients with AIS. They found that the Cronbach’s alpha values in all domains were greater than 0.7 [19]. Parent et al. also found that the English version of ISYQOL had good internal consistency (Cronbach’s alpha values ranging from 0.79 to 0.84) [49]. Additionally, ISYQOL-TC demonstrated excellent test–retest reliability for both the spine health and brace domains (all ICC values > 0.94) among participants with or without bracing. Canadian-French, Persian, and Polish versions of ISYQOL also showed excellent test–retest reliability (ICC > 0.9) [14,18,26], while the simplified Chinese and Arabic versions displayed moderate test–retest reliability (ICC > 0.7) [17,19].

The Rasch PCA of residuals supported the unidimensionality of ISYQOL-TC and indicated its distinct construct. As the original Italian version of ISYQOL is created based on the framework of Rasch analysis, ISYQOL is known to be more unidimensional and reliable than SRS-22r, which remains unsatisfactory according to the item response theory [15]. The raw variance explained by the measures of ISYQOL-TC exceeded 40% of the total variance. The first contrast of the residual (Second Major Component) eigenvalue of < 3 supported the unidimensionality of ISYQOL-TC. The infit and outfit statics of ISYQOL-TC revealed that all 20 items had high accuracy. Our study is the first study to use the Rasch PCA method in the Chinese community to demonstrate the unidimensionality of a translated version of ISYQOL, which agreed with the findings of the original ISYQOL (mean square infit and outfit between 0.5–1.5 and eigenvalue < 2) [15].

Interestingly, a recent study evaluated the measurement properties of six versions of translated ISYQOL questionnaires (i.e., English, Canadian-French, Spanish, Greek, Polish, and Turkish) using Rasch analysis, and found that four items related to positive thoughts about the spine poorly fitted the model of Rasch [50]. Therefore, a new 16-item ISYQOL International questionnaire was developed. Their differential item functioning analysis also found that seven out of the sixteen items were slightly affected by nationality. Although speculative, the good agreement between ISYQOL-TC and the original ISYQOL may be attributed to our adoption of the FACIT translation and linguistic validation approach, as well as the involvement of the developer of the original ISYQOL, which allowed a more accurate translation of convoluted Italian words or colloquialisms. Future studies should use the differential item functioning analysis to determine whether ISYQOL-TC has psychometric equivalence to the original ISYQOL or ISYQOL International questionnaire, which will guide the necessity of modifying the calibrations of items in ISYQOL-TC to enable comparisons of HRQOL in patients with AIS across different cultures.

### 4.2. Comparisons with SRS22r and Other Questionnaires

The ISYQOL-TC showed a strong positive correlation with total scores and the five subscale scores of the Chinese SRS-22r. Prior studies also demonstrated moderate correlations between the Canadian-French, simplified Chinese, or the Italian versions of ISYQOL interval scores and SRS-22r scores (ρ were 0.56, 0.62, and 0.71, respectively) [14,15,19]. Meanwhile, prior prospective case series and a systemic review reported that SRS-22r had ceiling effects between 47% and 52.9% in the functional domain [12,13,14], and high ceiling effects on the total scores and several domain scores [15,16]; the current study also revealed such an effect. Our participants’ mean SRS-22r score and standard deviation were 94.62 ± 8.84 out of 100, whereas those of ISYQOL-TC were 61.25% ± 15.60% (Table A8). Our findings indicate that the ISYQOL-TC is better than the Chinese SRS-22r in detecting improvements in HRQOL among patients with AIS, which concur with the findings of the English version [16,51]. Although the current study found no ceiling effect in ISYQOL-TC, the sensitivity of ISYQOL-TC in measuring post-treatment changes in HRQOL remains uncertain. Given that the responsiveness of the original or various translated versions of ISYQOL in monitoring changes in HRQOL among patients with AIS is unknown, future research should evaluate the responsiveness of ISYQOL-TC and its temporal measurement properties’ invariance.

In addition to overcoming the ceiling effect of SRS-22r [14,17,18,26], ISYQOL has other advantages. Compared to SRS-22r, ISYQOL has less questions, which allows users to complete the questionnaire in less than 10 min. Importantly, ISYQOL is a Rasch-consistent questionnaire that allows researchers/clinicians to directly compare the HRQOL of AIS patients with and without bracing. Theoretically, a good and fundamental scale should show unidimensionality, additivity, and suitability for the target population [49]. The current study and prior research support the notion that ISYQOL satisfies the requirements of unidimensionality, additivity, and generalizability [15,16,25,52]. Therefore, ISYQOL-TC is a good disease-specific HRQOL questionnaire for patients with AIS whose HRQOL may be compromised by pain, poor self-image, or concerns about progression [53,54].

ISYQOL-TC interval scores were moderately and negatively correlated with NPRS and GAD-7 scores. It is known that AIS patients with pain have a poorer HRQOL because pain may negatively impact their self-perceived health [55]. Because patients with AIS are more likely to experience back pain or sports-related injuries [5,6], it is not surprising to find a moderate negative correlation between ISYQOL-TC and NPRS scores in these patients. Additionally, the presence of AIS may negatively affect their self-image, which may give rise to anxiety in peer interaction. Our finding substantiates the notion that anxiety has a negative impact on one’s HRQOL [56,57].

Although patients with chronic diseases (e.g., multiple sclerosis) who adopted adaptive coping strategies showed better HRQOL [58], the current study found no significant correlation between the coping style and ISYQOL-TC scores. This inconsistency might be attributed to the fact that multiple factors might affect a person’s coping strategies (e.g., socioeconomic status, as well as family and social support). Therefore, it is difficult to accurately quantify the correlation between the coping style and HRQOL without considering various confounders [59].

The current study had some limitations. First, the data were collected from one scoliosis clinic, which affected the generalizability. However, this is because there are only two scoliosis clinics in Hong Kong, and so our participants should be a good representative sample. Second, the follow-up questionnaire was sent to participants by email. The participants might have been influenced by their parents/guardians when completing the questionnaire. However, our analysis showed high test–retest reliability, indicating that the influences from parents/guardians should be minimal.

### 4.3. Implications

Because the prevalence of AIS and its impacts on teenagers may be similar across ethnicities [60,61,62], it is essential to use a standardized and cross-culturally adapted disease-specific HRQOL questionnaire to compare the HRQOL of these patients across countries/regions. The cultural adaptation of the ISYQOL-TC allows teenagers reading traditional Chinese to use ISYQOL for HRQOL assessments. Clinicians can use ISYQOL-TC to better understand the impacts of AIS or bracing on these patients’ HRQOL, which may help to guide their clinical decision making. Importantly, it enables comparisons of AIS research findings from traditional Chinese users with users using other languages. The future field testing of ISYQOL-TC in evaluating the HRQOL of patients waiting for spine surgery or with a history of spine surgery is warranted. Further studies should also explore whether ISYQOL-TC can be used to evaluate HRQOL in patients with other spinal problems (e.g., degenerative scoliosis or kyphosis).

## 5. Conclusions

The 20-item ISYQOL-TC quantifies HRQOL in patients with AIS in Hong Kong. All 20 items in the ISYQOL-TC showed unidimensionality, as well as good internal consistency, content validity, structural validity, and construct validity. ISYQOL-TC showed no ceiling effect. Future studies should examine whether the ISYQOL-TC is responsive to changes in HRQOL among patients with AIS.

## Figures and Tables

**Figure 1 healthcare-11-02683-f001:**
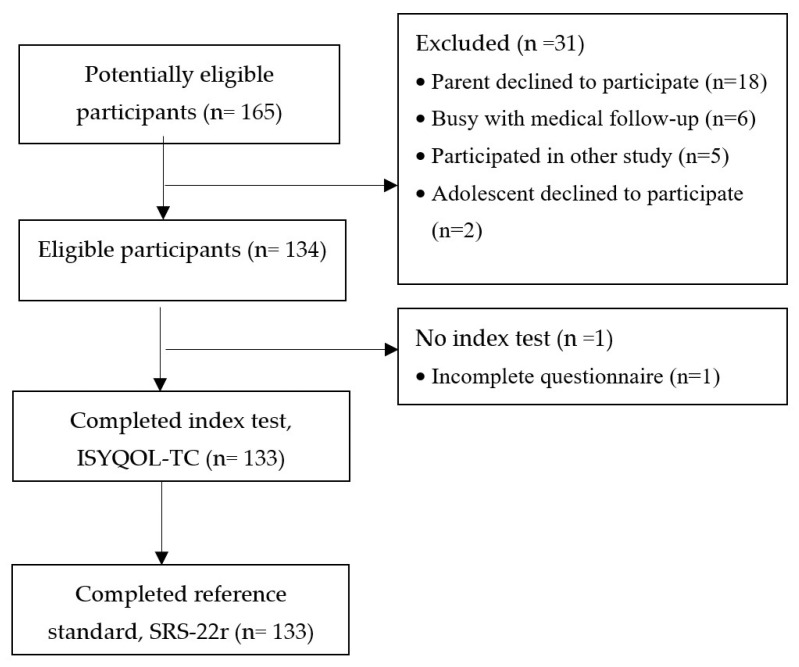
Study flowchart. Legend: ISYQOL-TC—Traditional Chinese version of Italian Spine Youth Quality of Life questionnaire, and SRS-22r—Scoliosis Research Society-22 revised questionnaire.

**Table 1 healthcare-11-02683-t001:** Demographics and clinical data of participants. (N = 133).

Number of Participants	Gender N (%)	Mean Age ±SD (years)	Curve Classification N (%)	Mean Curve Degree ±SD (°)
Brace (*n* = 57)	F = 48 (84%) M = 9 (16%)	13.5 ± 1.9	T: 1 (2%) TL: 13 (23%) L: 2 (4%) 2-curve: 33 (58%) 3-curve: 8 (14%)	28 ± 9
No brace (*n* = 76)	F = 51 (67%) M = 25 (33%)	14.2 ± 2.1	T: 10 (13%) TL: 21 (27.6%) L: 1 (1.3%) 2-curve: 39 (51.3%) 3-curve: 5 (6.6%)	26 ± 12
Total (*n* = 133)	F = 99 (74%) M = 34 (25%)	13.9 ± 2.1	T: 11 (8%) TL: 34 (26%) L: 3 (2%) 2-curve: 72 (54%) 3-curve: 13 (10%)	27 ± 11

Legend: SD—standard deviation; F—female; M—male; T—thoracic; TL—thoracolumbar; and L—lumbar.

**Table 2 healthcare-11-02683-t002:** Summary of Internal Consistency, Intraclass Correlation Coefficient, standard deviation, standard error of measurement, and 95% confidence minimal detectable change for traditional Chinese version of Italian Spine Youth Quality of Life questionnaire.

Questionnaire	Cronbach’s Alpha	ICC (95% Confidence Interval)	Mean ± SD (Raw Scores)	Mean ± SD (%)	SEM	MDC_95_
ISYOQL-TC (No brace) *n* = 76	0.90 **	0.95 (0.92–0.97) **	8 ± 6	63 ± 18	4%	12%
ISYOQL-TC (Brace) *n* = 57	0.89 **	0.96 (0.94–0.98) **	13 ± 7	59 ± 11	2%	6%
Spine health domain *n* = 133	0.89 **	0.95 (0.93–0.96) **	8 ± 5		1	3
Brace domain *n* = 57	0.79 **	0.96 (0.93–0.98) **	5 ± 3		1	1

Legend: ICC—intraclass correlation coefficient; SD—standard deviation; SEM—standard error of measurement; MDC95—95% confidence minimal detectable change; and ISYQOL-TC—traditional Chinese version of Italian Spine Youth Quality of Life questionnaire. ** Denotes statistical significance at *p* < 0.01.

**Table 3 healthcare-11-02683-t003:** Spearman Correlation between the interval measure of traditional Chinese version Italian Spine Youth Quality of Life questionnaire and Scoliosis Research Society-22 revised questionnaire scores.

SRS-22r Domains	ISYQOL-TC Interval Measure (%)			
	Mean ± SD	No Brace	Brace	Total
Function	24 ± 2	0.34 **	0.41 **	0.38 **
Pain	23 ± 2	0.41 **	0.33 **	0.41 **
Self-image	19 ± 3	0.58 **	0.64 **	0.59 **
Mental health	21 ± 3	0.55 **	0.56 **	0.55 **
Satisfaction	8 ± 1	0.24 **	0.24 **	0.26 **
Total	95 ± 9	0.63 **	0.67 **	0.65 **

Legend: SRS-22r—Scoliosis Research Society-22 revised questionnaire; and ISYQOL-TC—traditional Chinese version of Italian Spine Youth Quality of Life questionnaire. ** Denotes statistical significance at *p* < 0.01.

**Table 4 healthcare-11-02683-t004:** Correlation between the interval measure of traditional Chinese version of Italian Spine Youth Quality of Life questionnaire and the Simplified Coping Style Questionnaire, Patient Health Questionnaire-9, General Anxiety Disorder-7, the Mastery Scale, and Numeric Pain Rating Scale.

		ISYQOL-TC Interval Measure (%)		
	Mean ± SD	No Brace	Brace	Total
SCSQ-A	19 ± 7	0.11	0.10	0.103
SCSQ-P	9 ± 5	−0.14	−0.13	−0.135
PHQ-9	4 ± 4	−0.43 **	−0.41 **	−0.46 **
GAD-7	3 ± 4	−0.40 **	−0.43 **	−0.43 **
MAS	16 ± 5	0.43 **	0.43 **	0.44 **
NPRS	1 ± 2	−0.38 **	−0.41 **	−0.39 **

Legend: ISYQOL-TC—traditional Chinese version of Italian Spine Youth Quality of Life questionnaire; SCSQ-A—Simplified Coping Style Questionnaire-Active; SCSQ-P—Simplified Coping Style Questionnaire-Passive; PHQ-9—Patient Health Questionnaire-9; GAD-7—General Anxiety Disorder-7; MAS—Mastery Scale; and NPRS—Numeric Pain Rating Scale. ** Denotes statistical significance at *p* < 0.01.

## Data Availability

The data are available upon reasonable request.

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
