# Peer review of "Cross-Cultural Adaptation and Psychometric Properties of the Traditional Chinese Version of the Italian Spine Youth Quality of Life (ISYQOL) Questionnaire"

_healthcare, 2023, doi:10.3390/healthcare11192683_

Round 1

Reviewer 1 Report

Thank you for submitting your manuscript on the conversion and validation of the Italian Youth Spine Quality of Life (IYSQOL) with traditional Chinese. (IYSQOL-TC). The methodology was not familiar in achieving this validation. The main conclusion is the ability for Chinese surgeons to use the IYSQOL-TC in monitoring the treatment of their patients and in performing clinical research. I only have a few questions to be addressed in a revision.

(1). Cobb is not an angle rather a technique used to measure radiographic spinal angles (scoliosis, kyphosis, lordosis. etc). Cobb angle is a common term but is jargon and not appropriate for scientific studies. The correct term for scoliosis is "major coronal curve" or just "major curve". Please choose one and use it throughout your manuscript including any tables or figure legends where it appears.

Another jargon you have used is the term "x-ray". The appropriate term is "radiographic". Please make this change as well.

(2). It usually not appropriate to use decimal places in radiographic spinal measurements. The standard error of measurement is 3-5 degrees. Thus, the use of decimal places does not add accuracy or significance. Please round these to the nearest whole numbers throughout your manuscript again including all tables and figure legends. I do not think this will change any of your statistics, but they will need to be verified.

(3). The final concern is similar to the others. Percentages also do not typically need decimal places when the numerator is less than 200. Their use does not add accuracy or significance. Please round to the nearest whole number. 

Please verify that your other numerical results require truly one or two decimal places.

Author Response

We would like to thank the Reviewer for the support and constructive comments. We have attached our response in the attached file and revised the manuscript accordingly. Please see the attachment. 

Reviewer 2 Report

Thank you very much for the opportunity to review the manuscript titled "Cross-cultural Adaptation and Psychometric Properties of the Traditional Chinese Version of the Italian Spine Youth Quality of Life (ISYQOL) Questionnaire". In my opinion, the work has been carried out with great methodological rigor, although the version of the manuscript being evaluated presents some opportunities for improvement.

For keywords, it seems more appropriate to me to choose mesh terms that can facilitate the search for this document in the future. For example "scoliosis" instead of "Adolescent Idiopathic Scoliosis".

Introduction: There are several words that begin with a capital letter after a comma. Please correct.

The expression "teenagers" with AIS may be redundant. Better to substitute the term "patients" whenever possible.

In the material and methods section, subsection 2.5.3. on psychometric properties should include the following, which could be numbered as 2.5.3.1, 2.5.3.2.....

The cross-cultural adaptation process seems well carried out and the psychometric properties correctly evaluated. My concern is why exploratory factor analysis has not been performed for structural validity analysis?

In the cross-cultural adaptation to different languages of Europe and America (Negrini S, Zaina F, Buyukaslan A, Fortin C, Karavidas N, Kotwicki T, et al. Cross-cultural validation of the Italian Spine Youth Quality of Life questionnaire: the ISYQOL international Eur J Phys Rehabil Med 2023;59:364-76), this questionnaire suffered the elimination of several items until it remained at 16. How do the authors assess that the Chinese version maintains the original 20 items?

One of the limitations of the study is the lack of a sensitivity analysis to change of the evaluated questionnaire. I would like the authors to discuss whether this psychometric property has been previously evaluated in any of the versions of the study and also include this issue as one of the limitations of the study.

Author Response

Thank you the suggestions. We have provided a point-by-point response to the Reviewer's comments. Please see the attachment. Thank you very much! 

Reviewer 3 Report

Overall, manuscript has merit, and I recommend publication after major revisions. The writing and structure can be improved. For example, the Data Analysis section could be summarized more as opposed to having this list structure. But it may just be personal preference. Please see more detailed comments below:

Introduction:

The manuscript provides a detailed background on Adolescent Idiopathic Scoliosis (AIS), emphasizing its significance as the most prevalent three-dimensional structural spine change among teenagers aged between 10 and 18 years. It underscores the importance of early detection and intervention. Two key instruments for assessing Health-Related Quality of Life (HRQOL) in AIS patients, the Scoliosis Research Society-22 Revised Questionnaire (SRS-22r) and the Italian Spine Youth Quality of Life questionnaire (ISYQOL), are introduced. While the SRS-22r is widely used, it has limitations, notably the high ceiling effects in the pain domain. The manuscript suggests the ISYQOL as a potentially superior alternative.

—In the sentence starting with "Adolescent Idiopathic scoliosis", replace "scoliosis" with "Scoliosis" to ensure appropriate capitalization at the beginning of a sentence.

—In the sentence starting with "Adolescent Idiopathic Scoliosis (AIS)", the numeral "3-dimensional" should be written as "three-dimensional" for consistency and clarity.

—In the sentence starting with "SRS-22r has been translated", replace "and is" with "and it is" to ensure coherence.

—In the sentence starting with "Researchers/clinicians can directly", it might be clearer to say "Researchers and clinicians" instead of using a slash.

—In the final paragraph, while the purpose of the study is presented clearly, it might be clearer to rephrase "Given the above" to something like "Considering the aforementioned background".

Methods:

—In the sentence starting with "This study was approval", replace "approval" with "approved" to correct the verb tense.

—In the section title "2.1. Study design", consider using consistent capitalization for section titles. For example, "Study Design" instead of "Study design".

—The sentence starting with "Cantonese speaking teenagers aged" might be clearer if rephrased to "Teenagers, who spoke Cantonese and were aged between 10 and 18 years,".

—In the sentence "Participants needed to have a thoracic", consider replacing "needed" with "were required" for clarity.

—In the sentence starting with "Teenagers were excluded if they had", it might be clearer to use bullet points for the exclusion criteria.

—In the sentence "Six teenagers with AIS participated", consider specifying that these are different from the main study participants, if that's the case.

—The sentence "The sample size for the validation study" is quite dense and might benefit from breaking into smaller sentences or using bullet points for clarity.

—In the section "2.3. Phase 1: Translation and cross-cultural adaptation of questionnaire", consider using consistent capitalization: "Phase 1: Translation and Cross-cultural Adaptation of Questionnaire".

—In the sentence "The translation process followed the FACIT translation method", consider briefly defining or describing the FACIT translation method for readers who might not be familiar with it.

—It would be useful to present the items of the questionnaire.

Method/Results:

There is insufficient detail about the Rasch analysis approach taken and the results. There was no mention of differential item functioning, for example.

Discussion:

The authors did an admirable job drawing parallels between their results and previous studies, particularly in terms of internal consistency and test-retest reliability. Noting similarities with findings from other translated versions, such as the simplified Chinese and English iterations, provides a broader context.

—While the authors provided a broad comparison of their findings with existing literature, a deeper exploration of why certain results aligned or differed from previous studies might add more depth.

—It would be beneficial to see more about potential applications for the ISYQOL-TC, especially given its demonstrated efficacy.

Only minor editing recommended.

Author Response

Thank you for taking the time to review our manuscript. We have provided the point-to-point response in the attached file. Please see the attachment. 
